# Human-AI Collaborative Bayesian Optimisation

**Arun Kumar A V, Santu Rana, Alistair Shilton, Svetha Venkatesh**
Applied Artificial Intelligence Institute ($A^2I^2$), Deakin University
Waurn Ponds, Geelong, Australia
`{aanjanapuravenk,santu.rana,alistair.shilton,`
`svetha.venkatesh}@deakin.edu.au`

## Abstract

Human-AI collaboration looks at harnessing the complementary strengths of both humans and AI. We propose a new method for human-AI collaboration in Bayesian optimisation where the optimum is mainly pursued by the Bayesian optimisation algorithm following complex computation, whilst getting occasional help from the accompanying expert having a deeper knowledge of the underlying physical phenomenon. We expect experts to have some understanding of the correlation structures of the experimental system, but not the location of the optimum. The expert provides feedback by either changing the current recommendation or providing her belief on the good and bad regions of the search space based on the current observations. Our proposed method takes such feedback to build a model that aligns with the expert's model and then uses it for optimisation. We provide theoretical underpinning on why such an approach may be more efficient than the one without expert's feedback. The empirical results show the robustness and superiority of our method with promising efficiency gains.

## 1 Introduction

The deep penetration of AI systems into wider society demonstrates its ability to deal with complex real-world tasks. However, there still exists a plethora of complicated real-world problems [Roccetti et al., 2020, Hechler et al., 2020] that AI systems are unable to properly address. In such complex real-world tasks, human-AI collaborative systems can provide a viable alternative. Through appropriate collaboration strategy we can supplement the speed, scalability and quantitative abilities of AI systems with the reasoning and abstraction ability of human users and make the combination exceedingly capable [Carroll et al., 2019, Maadi et al., 2021].

Recent studies, mostly proposed in supervised learning setting, used either AI requesting help of human experts if its prediction lacks confidence [Madras et al., 2017] or as guided by a meta-policy that focuses on the joint performance [Wilder et al., 2020]. Mozannar and Sontag [2020], De et al. [2020] considered also the cost associated with human decision making while learning a joint policy. Bayesian optimisation (BO) [Brochu et al., 2010], an efficient tool for global optimisation that can potentially play a much bigger role in scientific experimentation and product design, provides a rich playground for human-AI collaboration paradigm because the nature of applications ensures that there would be an expert running the optimisation process. There are some works on incorporating expert's knowledge in Bayesian optimisation procedure. However, nearly all of the work relies on expert articulating a static type of knowledge in the beginning of the optimisation process either in terms of trends [Li et al., 2018, Riihimäki and Vehtari, 2010], shapes of functions [Andersen et al., 2017, Venkatesh et al., 2019] or suggesting a similar experimental system for transfer learning [Joy et al., 2016, Shilton et al., 2017]. In reality, most of the time experts may not be able to reveal knowledge in terms of such explicit specifications. Expert's knowledge will also more likely evolve over the course of optimisation process through assimilation of the new data points. *Unfortunately,*

36th Conference on Neural Information Processing Systems (NeurIPS 2022).

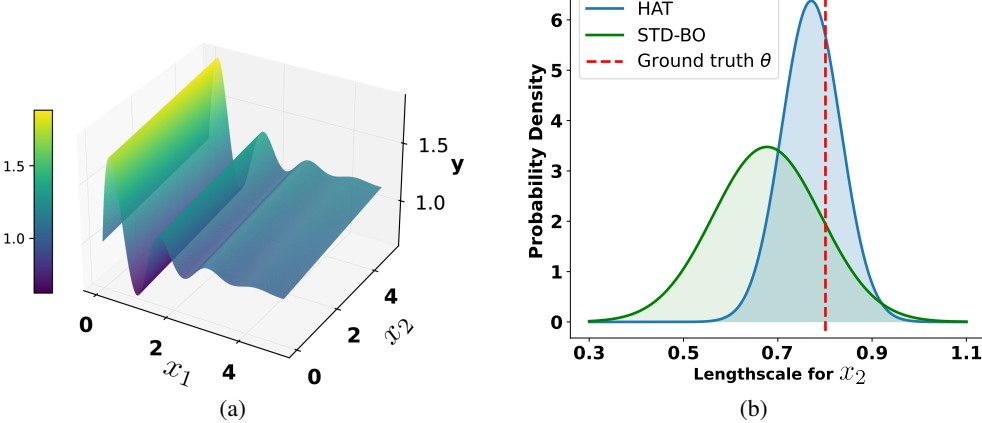

(a)    (b)

Figure 1: (a) Damped oscillator function (b) Hyperparameter distribution obtained using our proposed HAT based BO (blue) and standard BO (green). Red line shows the ground truth lengthscale.

*a true collaborative system that takes into account such a fluid knowledge from the human expert via an intuitive communication interface is so far lacking.*

To address this problem, we propose a new **H**uman-**AI T**eaming (HAT) based Bayesian optimisation framework that takes expert input during the course of optimisation through one of the two ways: **(i)** correction of the current recommendation, or **(ii)** specification of a good region and bad region based on the current observation set. We believe these are easier to specify, than say, a global trend or global shape of a function because an expert may be able to provide reasoning surrounding a small neighbourhood of a sample point. Once such feedback is given, we incorporate it into Bayesian optimisation by changing the model selection process that ensures that such relative knowledge that the corrected recommendation is a more promising one compared to the current recommendation or the good region is more promising than the bad region are maintained. We use the acquisition function as the means to compute the *promise* and use the aforesaid feedback during model selection via maximum likelihood. The model derived from these constraints is subsequently used for the next recommendation. Our framework allows an expert to intervene only when they are willing to intervene and also allows them to evolve their knowledge over the course of optimisation.

Theoretically, we show that when Gaussian process is used to model the function, this extra channel of information narrows the space of functions. We characterise such narrower distributions in terms of Sobolev spaces [Tartar, 2007] and show how it admits tighter upper bound of the total regret when an appropriate trade-off factor is used in the acquisition function. Further, we show in Figure 1a and 1b, a case of HAT framework vs standard likelihood based model selection of lengthscales of a Squared Exponential (SE) kernel. For a particular dimension $(x_2)$ we observe that the HAT framework with an expert having ground truth knowledge infers models that are distributed more narrowly than the standard one, inducing a smaller function space. We evaluate the performance of our proposed methods on both synthetic benchmark functions and real-world datasets. We tune the hyperparameters of the Support Vector Machine (SVM) classifier [Burges, 1998, Christmann and Steinwart, 2008] operating on publicly available multi-dimensional real-world datasets. We compare the optimisation performance of our proposed Human-AI Teaming (HAT) framework against different variations of the Bayesian optimisation algorithm. The empirical results prove the efficacy of our proposed framework. We also provide a small-scale real-user study to demonstrate the superiority of our proposed Human-AI Teaming approach over AI only approach.

## 2   Background

**Notations.** $\mathbb{R}$ for Reals. $\mathbb{X}$ is an index set and $\mathbf{x} \in \mathbb{X}$. $\mathbb{N}_n = \{1, 2, \cdots, n\}$. $\cup$ for the set union. $|\cdot|$ is the determinant. We use lower case bold fonts $\mathbf{v}$ for vectors. $v_i$ for $i^{\text{th}}$ element in a vector $\mathbf{v}$. $\mathbf{v}^{\mathsf{T}}$ for the transpose of a vector $\mathbf{v}$. We use upper case bold fonts $\mathbf{M}$ (and bold Greek symbols) for matrices and $M_{ij}$ for the element in $i^{\text{th}}$row and $j^{\text{th}}$ column of $\mathbf{M}$. abs$(\cdot)$ for the absolute value.

## 2.1 Gaussian Process

Gaussian Process (GP) [Williams and Rasmussen, 2006] is a non-parametric model, that provides a flexible framework for placing prior on functions. GP defines a distribution over the set of possible functions given observations. Though there are other popular surrogate models such as Wiener process [Kushner, 1964] and Student-t process [Shah et al., 2014], GP is still the preferred model because of its simplicity. A GP is completely specified by a mean function $\mu(\mathbf{x})$ and a kernel function $k(\mathbf{x}, \mathbf{x}')$. The unknown objective function $f(\mathbf{x})$ is modelled using GP as $f(\mathbf{x}) \sim \mathcal{GP}(\mu(\mathbf{x}), k(\mathbf{x}, \mathbf{x}'))$. If $\mathcal{D}_{1:t} = \{\mathbf{x}_{1:t}, \mathbf{y}_{1:t}\}$ denotes the set of $t$ observations, then according to the properties of the Gaussian process, a new observation $(\mathbf{x}_{t+1}, y_{t+1})$ and the set of previously observed data samples $\mathcal{D}_{1:t}$ are jointly Gaussian. Therefore, the posterior distribution for the new observation $y_{t+1}$ is computed as $\mathcal{P}(y_{t+1}|\mathcal{D}_{1:t}, \mathbf{x}_{t+1}) = \mathcal{N}(\mu(\mathbf{x}_{t+1}), \sigma^2(\mathbf{x}_{t+1}))$, where $\mu(\mathbf{x}_{t+1}) = \mathbf{k}^\intercal[\mathbf{K} + \sigma^2_{\mathrm{GN}}\mathbf{I}]^{-1}\mathbf{y}$, $\sigma^2(\mathbf{x}_{t+1}) = k(\mathbf{x}_{t+1}, \mathbf{x}_{t+1}) - \mathbf{k}^\intercal[\mathbf{K} + \sigma^2_{\mathrm{GN}}\mathbf{I}]^{-1}\mathbf{k}$, $\mathbf{k} = [k(\mathbf{x}_1, \mathbf{x}_{t+1}), \cdots, k(\mathbf{x}_t, \mathbf{x}_{t+1})]$, $K_{ij} = k(\mathbf{x}_i, \mathbf{x}_j) \; \forall i, \forall j \in \mathbb{N}_t$ and $\sigma^2_{\mathrm{GN}}$ is the variance of the white Gaussian noise.

The kernel (covariance) function $k : \mathbb{X} \times \mathbb{X} \to \mathbb{R}$ used in Gaussian process plays a crucial role in modelling the Gaussian process surrogates. The kernel function incorporates our prior belief about the unknown objective function $f$. The kernel hyperparameters $\theta$ are usually estimated by maximising the marginal likelihood given as $\mathcal{L} = p(\mathbf{y} \mid X, \theta) = \int p(\mathbf{y} \mid \mathbf{f}) \, p(\mathbf{f} \mid X, \theta) \, d\mathbf{f}$. The closed-form formulation of Gaussian process log-likelihood is given as:

$$\log \mathcal{L} = -\frac{1}{2}(\mathbf{y}^\intercal(\mathbf{K} + \sigma^2_{\mathrm{GN}}\mathbf{I})^{-1}\mathbf{y}) - \frac{1}{2}\log \mid \mathbf{K} + \sigma^2_{\mathrm{GN}}\mathbf{I} \mid -\frac{t}{2}\log(2\pi) \tag{1}$$

## 2.2 Bayesian Optimisation

Bayesian optimisation (BO) [Shahriari et al., 2015, Frazier, 2018] has proliferated into many domains, including sensor networks [Garnett et al., 2010, Zhang et al., 2018], robotics [Martinez-Cantin et al., 2007, Bossens and Tarapore, 2021], intelligent environmental monitoring [Marchant and Ramos, 2012], and information retrieval [Wang et al., 2014]. BO offers an efficient framework for the global optimisation of an expensive and noisy black-box function $f(\mathbf{x})$, represented as:

$$\mathbf{x}^* = \mathrm{argmax}_{\mathbf{x} \in \mathbb{X}} f(\mathbf{x}) \tag{2}$$

where $\mathbb{X}$ corresponds to a compact search space. The observations of the objective function $f(\mathbf{x})$ are assumed to be corrupted with a white Gaussian noise *i.e.*, $y_t = f(\mathbf{x}_t) + \epsilon_t$, where $\epsilon_t \sim \mathcal{N}(0, \sigma^2_{\mathrm{GN}})$. Bayesian optimisation models the objective function $f$ by placing a prior distribution over the space of unknown objective functions and then use it to determine the next best locations to sample. Specifically it consists of two main components: **(i)** a Gaussian process model [Williams and Rasmussen, 2006], and **(ii)** an acquisition function [Kushner, 1964, Wilson et al., 2018]. The GP predictive distribution obtained is used to select the next sampling location with the high promise of finding the optima. The acquisition function guides the search for optima by suitably balancing the exploration-exploitation trade-off. Srinivas et al. [2012] have discussed the Gaussian Process-Upper Confidence Bound (GP-UCB) acquisition function using the upper confidence bound selection criterion. A GP-UCB acquisition function at $t^{\mathrm{th}}$ iteration is given by:

$$\alpha_{\mathrm{GP\text{-}UCB}}(\mathbf{x}) = \mu(\mathbf{x}) + \sqrt{\beta_t}\,\sigma(\mathbf{x}) \tag{3}$$

where $\beta_t$ is a trade-off parameter that balances between the exploitation and exploration. Chowdhury and Gopalan [2017] have discussed the possible choices for $\beta_t$ and their implications on the overall regret. In general, to achieve a faster convergence rate $\beta_t$ is increased with $O(\log t)$. An iterative algorithm for the standard Bayesian optimisation is provided in the supplementary material (A.1).

## 2.3 Reproducing Kernel Hilbert Spaces (RKHS)

The kernel function used in GP surrogate modelling is associated with a unique Reproducing Kernel Hilbert Space (RKHS). Let $k$ be a symmetric positive definite kernel on $\mathbb{R}^d$ and set $k_\mathbf{x}(t) = k(t - \mathbf{x})$. For a given compact space $\mathbb{X} \subseteq \mathbb{R}^d$, let $\phi(\mathbb{X})$ be the space of functions with the mapping $\mathbb{X} \to \mathbb{R}$, spanned by the kernel $k_\mathbf{x}$. If $\phi(\mathbb{X})$ is provided with the inner product represented as $\langle k_\mathbf{x}, k_{\mathbf{x}'} \rangle = k(\mathbf{x} - \mathbf{x}')$, then the completion of $\phi(\mathbf{x})$ is the reproducing kernel Hilbert space $\mathcal{H}_k(\mathbb{X})$ of $k$ on $\mathbb{X}$.

## 3   Framework

We propose a Human-AI Teaming (HAT) based BO framework to involve the human experts in the optimisation loop to accelerate the optimisation by utilising their knowledge in the surrogate modelling. First, we discuss the additional knowledge accessible to human experts and then we provide two variants of the HAT framework differing mainly on how the expert knowledge is communicated. The optimisation problem is same as that of the standard BO described in the Background section.

### 3.1   Human Experimenter Knowledge

In many real-world applications, we can expect the human experts to have knowledge about the correlations structure in the underlying function but not about the shape of the function or about the location of the optima. With such knowledge, expert may be able to provide a better recommendation, at least in the beginning. We use those expert suggestions at different stages of the optimisation process to assess the deviations of the current AI model and correct them accordingly. The inherent knowledge about the correlations structure of the objective function can take the form of full or partial knowledge of the kernel function, *e.g.,* function is smoother along one dimension than the other. In either case, the expert can provide input regarding the relative difference between two locations in the search space in terms of their utility as the next sample point based on her cognitive model and the selection criteria. For example, if the expert knowledge of the kernel says that the objective function along one dimension is smoothly varying then an expert may be able to adjust the current recommendation by "pushing" that variable further in the direction that takes it away from the existing observations. If the expert has full knowledge of the kernel then she may be able to perform such rectifications along all the input variables. We also assume that the expert is running a form of Bayesian optimisation strategy as it has been demonstrated that human active search typically mimics Bayesian optimisation [Borji and Itti, 2013]. Next, we discuss two ways of giving feedback.

### 3.2   Rectifying Current Recommendation

In this approach, we propose to correct the previously suggested AI model's observations $\mathcal{X}^A$ in the light of human expert knowledge. The human expert knowledge is represented using a set of observations $\mathcal{X}^E$ generated using her cognitive model reflecting full or partial knowledge of the kernel function. At each iteration $t$, the kernel hyperparameters $\Theta^* \in \mathbb{R}^d$ of the AI model (*i.e.,* the BO process) are tuned by maximising the log-likelihood mentioned in Eq. (1). Using this tuned GP surrogate, the AI model suggests the next potential candidate $(\mathbf{x}_{t+1}^A)$ for the function evaluation. In the iterations ($t \in H$) that involve the expert rectification of the current AI recommendation, expert suggests the next potential candidate $(\mathbf{x}_{t+1}^E)$ based on her cognitive model. Then, based on such previous expert recommendations we add constraints on the model selection via acquisition functions ($u_{\text{GP-UCB}}$) such that the next potential candidate suggested by the AI model has to take into account the corrections enforced by the expert observations $\mathbf{x}_i^E \ \forall i \in H$ collected in $\mathcal{X}^E$. Here we assume that a human expert is also using a form of BO strategy but with a better model. We should note that the expert does not necessarily need to provide the optimal point according to the correct formulation. The expert needs to only provide a more promising location than what the machine recommended. Hence, the constraints take the form as acquisition function ($u_{\text{GP-UCB}}$) at $\mathbf{x}^E$ is better than the acquisition function at $\mathbf{x}^A$. The resulting constrained model selection problem is given as:

$$\Theta^* = \underset{\Theta}{\operatorname{argmax}} \ \log \mathcal{L} \quad \text{s.t} \quad u_{\text{GP-UCB}}(\mathbf{x}_i^E | \mathcal{D}_{-1}) > u_{\text{GP-UCB}}(\mathbf{x}_i^A | \mathcal{D}_{-1}) \ \forall \mathbf{x}_i^A \in \mathcal{X}^A, \ \mathbf{x}_i^E \in \mathcal{X}^E, \ i \in H \quad (4)$$

where $\mathcal{D}$ corresponds to the set of all observations $\{(\mathbf{x}, y = f(\mathbf{x}))\}$ accumulated by the AI model, $\mathbf{x}_i^A$ corresponds to the $i^{\text{th}}$ observation suggested by the AI model collected in $\mathcal{X}^A$. These constraints (collected in a set $\mathbb{C}$) are enforced in the model selection process in the next round. The intuition behind this constraint set is to restrict ourselves to the hyperparameter distribution that make the AI acquisition function more closely resemble the human acquisition function.

One may wonder **(i)** why a human expert would behave like a BO algorithm?, and **(ii)** why humans themselves cannot perform the optimisation?. To answer the former we refer to the work of Borji and Itti [2013], where they showed that when humans are presented with a search problem they often tend to behave like a BO strategy, *i.e.,* we tend to mix a bit of exploitation and exploration when sampling for the next point. Here we assume that the strategy is like the GP-UCB strategy.

---
**Algorithm 1** HAT - Rectifying Recommendations (HAT-RR)
---
**Input**: Initial observations $\mathcal{D}_{t'} = \{(\mathbf{x}_{1:t'}, y_{1:t'})\}$, Sampling budget $T$

1. Initialise expert suggestions $\mathcal{X}^E = \emptyset$, AI suggestions $\mathcal{X}^A = \emptyset$, Expert suggestion iterations $H = \emptyset$, Constraints list $\mathbb{C} = \emptyset$

2. **for** iterations $t = t', \cdots, T$ **do**

3.      Optimise $\Theta^* = \underset{\Theta}{\arg\max} \log \mathcal{L}$, such that constraints in $\mathbb{C}$ are satisfied

        **if** expert inputs are available *i.e.,* $\mathcal{X}^E \neq \emptyset$, **then**

            Add constraint $u_{\text{GP-UCB}}(\mathbf{x}_i^E | \mathcal{D}_{i-1}) > u_{\text{GP-UCB}}(\mathbf{x}_i^A | \mathcal{D}_{i-1})$ to $\mathbb{C}$

            $\forall \mathbf{x}_i^A \in \mathcal{X}^A, \mathbf{x}_i^E \in \mathcal{X}^E, \forall i \in H$

4.      Find the next query point $\mathbf{x}_{t+1}^A = \underset{\mathbb{X}}{\arg\max}\, u_{\text{GP-UCB}}(\mathbf{x})$

5.      $\mathcal{X}^A = \mathcal{X}^A \cup (\mathbf{x}_{t+1}^A)$

6.      **if** human expert wants to intervene, **then**

7.          Obtain expert input $\mathbf{x}_{t+1}^E$

8.          $\mathcal{X}^E = \mathcal{X}^E \cup (\mathbf{x}_{t+1}^E), \mathbf{x}_{t+1} = \mathbf{x}_{t+1}^E, H = H \cup \{t\}$

9.      **else** $\mathbf{x}_{t+1} = \mathbf{x}_{t+1}^A$

10.      Query the objective function $f(\mathbf{x})$ to find $y_{t+1} = f(\mathbf{x}_{t+1}) + \epsilon_{t+1}$

11.      Augment data $\mathcal{D}_{1:t+1} = \mathcal{D}_{1:t} \cup \{(\mathbf{x}_{t+1}, y_{t+1})\}$

12. **end for**
---

However, as we will see in the experiment, even if a human expert uses a noisy version of the GP-UCB strategy, our proposed method can still be robust to that. To answer the latter, we say that while human may possibly get to the optima, it may be extremely taxing for her to do so at every iteration, and also when the number of observations are high. Here, we do not need the expert to specify the optima of their acquisition function, but to suggest a better location than $\mathbf{x}^A$. We believe as humans we may be able to do that more easily than say running the actual BO which involves solving the global optimisation of the acquisition function. The complete procedure for Human-AI Teaming (HAT) with experts rectifying recommendations (HAT-RR) is given in Algorithm 1.

### 3.3 Good Regions vs Bad Regions

In contrast to the aforesaid approach, the expert can suggest a good region and a bad region to sample next. A good region is defined by a set of candidate points that promise high probability of finding the optima *i.e.,* a set of points that have similar utilities to be the next sampling location. Further, such good candidate points can be seen as the locations suggested by maximising expert's acquisition functions. On the other hand, a bad region corresponds to the portion of the input space that has very low probability to further improve the current solution. A bad region can be visualised as a set of points that have a very low utility to be the next sampling location as indicated by the low values of the acquisition function. To keep the presentation simple, we formulate our problem based on a good point and a bad point. The algorithm can be generalised to the case of region by adding constraint per pair of samples from those regions.

The expert suggestions about the good point $(\mathbf{x}^g)$ and the bad point $(\mathbf{x}^b)$ is used to fine-tune the AI model by replacing its current optimal hyperparameters $(\Theta^*)$ with a hyperparameter set $(\Theta^+)$ that is more aligned with the expert's cognitive model, in addition to achieving good likelihood. To accomplish this we construct a co-objective that maximises the value difference between the acquisition function of the good point and the bad point. The co-objective is mathematically represented as:

$$\Theta^+ = \underset{\text{start} \leftarrow \Theta^*}{\arg\max} \sum_{i \in H} (u_{\text{GP-UCB}}(\mathbf{x}_i^g | \mathcal{D}_{i-1}) - u_{\text{GP-UCB}}(\mathbf{x}_i^b | \mathcal{D}_{i-1})), \text{ s.t } (\log \mathcal{L})_{\Theta^+} \geq \Delta(\log \mathcal{L})_{\Theta^*} \quad (5)$$

where $\mathcal{D}$ corresponds to the set of all observations accumulated by the AI model. The bi-objective problem is solved in two steps: **(i)** first, the pure maximum likelihood (Eq. (4)) based hyperpa-

**Algorithm 2** Human-AI Teaming: Good Vs Bad Regions (HAT-DM)
___
**Input**: Initial observations $\mathcal{D}_{t'} = \{(\mathbf{x}_{1:t'}, y_{1:t'})\}$, Sampling budget $T$, Threshold $\Delta$

   1. Initialise expert suggestions $\mathcal{X}^E = \emptyset$, Human suggestion iterations $H = \emptyset$

   2. **for** iterations $t = t', \cdots, T$ **do**

   3.      **if** human expert wants to intervene, **then**

   4.          Obtain good and bad points $\mathbf{x}_t^g, \mathbf{x}_t^b$ from the expert

$$\mathcal{X}^E = \mathcal{X}^E \cup (\mathbf{x}_t^g, \mathbf{x}_t^b), H = H \cup \{t\}$$

   5.         Optimise hyperparameters: $\Theta^* = \underset{\Theta}{\operatorname{argmax}} \log \mathcal{L}$

   6.         **if** expert suggestions are available *i.e.,* $\mathcal{X}^E \neq \emptyset$, **then**

$$\Theta^+ = \underset{\text{start} \leftarrow \Theta^*}{\operatorname{argmax}} \sum_{i \in H} (u_{\text{GP-UCB}}(\mathbf{x}_i^g | \mathcal{D}_{i-1}) - u_{\text{GP-UCB}}(\mathbf{x}_i^b | \mathcal{D}_{i-1}))$$

such that $(\log \mathcal{L})_{\Theta^+} \geq \Delta (\log \mathcal{L})_{\Theta^*}$

   7.      Find the next query point $\mathbf{x}_{t+1} = \underset{\mathbb{X}}{\operatorname{argmax}} \, u_{\text{GP-UCB}}(\mathbf{x})$

   8.      Query the objective function $f(\mathbf{x})$ to find $y_{t+1} = f(\mathbf{x}_{t+1}) + \epsilon_{t+1}$

   9.      Augment data $\mathcal{D}_{1:t+1} = \mathcal{D}_{1:t} \cup \{(\mathbf{x}_{t+1}, y_{t+1})\}$

 10. **end for**
___

rameter estimation ($\Theta^* = \underset{\Theta}{\operatorname{argmax}} \log \mathcal{L}$) is performed, and then **(ii)** the co-objective (Eq. (5)) is solved by constraining the likelihood to be $\Delta$-close (threshold) to the likelihood obtained in the previous step. The overall algorithm for the HAT framework with the aforementioned difference maximisation strategy (HAT-DM) is provided in Algorithm 2.

## 4 Theoretical Analysis

Our theoretical analysis relies on the notion of Sobolev spaces [Tartar, 2007] and its extension to Hilbert spaces. We start with the preliminary results required for the theoretical analysis of our proposed method. Chowdhury and Gopalan [2017] have discussed the regret bounds of a variant of the GP-UCB algorithm in terms of the norm bounds defined on the objective function $f$.

**Lemma 1**: *Let $f : \mathbb{X} \to \mathbb{R}$, where $\mathbb{X} \subset \mathbb{R}^d$ and $d$ is the number of dimensions. Let $\mathcal{H}_k$ be the Reproducing Kernel Hilbert Space (RKHS) associated with the kernel function $k(\cdot, \cdot)$ such that the RKHS norm is bounded in the ball of radius $\mathcal{R}$ i.e., $\|f\|_k \leq \mathcal{B}_{\mathcal{R}}$. Then there exists a sequence of trade-off factors $\beta_t(\mathcal{R}, \delta)$ such that it holds with probability at least $1 - \delta$ that $|f(\mathbf{x}) - \mu_t(\mathbf{x})| \leq \sqrt{\beta_t}\sigma_t(\mathbf{x}), \forall \mathbf{x} \in \mathbb{X}$.*

Lemma 1 states that $f$ is highly probable to be contained in the confidence intervals induced by the GP predictions ($\mu_t(\cdot)$ and $\sigma_t(\cdot)$) using kernel $k : \mathbb{X} \times \mathbb{X} \to \mathbb{R}$ with the appropriate scaling induced by $\beta_t$ as discussed in Chowdhury and Gopalan [2017]. As $k$ defines the size of the function space, the RKHS norm $\|f\|_k = \sqrt{\langle f, f \rangle_k}$ measures the complexity of functions $f \in \mathcal{H}_k$. By considering larger norm-balls, we account for more complex functions in the given input space.

In the Human-AI Teaming (HAT) framework, the kernel hyperparameters $\Theta_{\text{HAT}}$ are tuned in the light of additional channel of information provided by the human experts. The expert suggestions incorporated in the modelling process have a significant impact on the resultant hyperparameter distribution (see Figure 1a and 1b) and thus, the latent function space. Our HAT framework disregards some portions of the hyperparameter distribution that violate the constraints added in the modelling process and restricts itself to the (narrower) feasible regions. Further, the search for a better hyperparameter set that is more aligned with the expert's cognitive model in the vicinity of the optimal hyperparameter set obtained by maximising the log-likelihood also results in a narrower

distribution (governed by the threshold $\Delta$). Thus, our proposed framework deals with a narrower distribution of functions controlled by the modelling constraints enforced in the optimisation process. Such narrower distributions are spanned by the RKHS $\mathcal{H}_{k_{\Theta_{\mathrm{HAT}}}}$. We characterise the RKHS $\mathcal{H}_{k_{\Theta_{\mathrm{HAT}}}}$ of our framework using the Sobolev spaces [Tartar, 2007]. Sobolev Hilbert space $\mathcal{H}^s$ is a special case of Sobolev spaces obtained by restricting the RKHS associated with the kernel function $k_\Theta$. With mild regularity assumptions on the boundary conditions and the smoothness of Sobolev Hilbert spaces, we draw similarities between the underlying constraints in Sobolev Hilbert spaces and the constraints enforced to construct RKHS $\mathcal{H}_{k_{\Theta_{\mathrm{HAT}}}}$ in the HAT framework. We use the mathematical formulations of the RKHS norm in Sobolev Hilbert spaces to characterise the RKHS norm of the HAT framework. We refer to the supplementary material for the definition of Sobolev spaces. The following theorem establishes the relation between the RKHS norms of the HAT framework and the standard Bayesian optimisation framework.

**Theorem 1:** *Let $\mathcal{H}(\mathbb{R}^d)$ be the space of real continuous functions $f_k \in L^2(\mathbb{R}^d)$. Let $k_{\Theta_{\mathrm{HAT}}}$ be the kernel function with hyperparameters $\Theta_{\mathrm{HAT}}$ used in the HAT framework. Let $\mathcal{B}_{\mathcal{R}_{\mathrm{HAT}}}$ be the norm-ball of radius $\mathcal{R}_{\mathrm{HAT}}$ induced in HAT based BO and let $\mathcal{B}_{\mathcal{R}_{\mathrm{STD}}}$ be the norm-ball for the standard BO. If $\mathcal{H}_{k_{\Theta_{\mathrm{HAT}}}}(\mathbb{X})$ is the reproducing Hilbert space of functions $f_k = g|_{\mathbb{X}}$ associated with the kernel $k_{\Theta_{\mathrm{HAT}}}$, then with high certainty it holds $\|f\|_{k_{\mathrm{HAT}}} \leq \mathcal{B}_{\mathcal{R}_{\mathrm{HAT}}}$ and $\mathcal{B}_{\mathcal{R}_{\mathrm{HAT}}} \leq \mathcal{B}_{\mathcal{R}_{\mathrm{STD}}}$.*

The proof of Theorem 1 is provided in the supplementary material. With the results established in Theorem 1, we conclude that the additional channel of human experimenter knowledge enforced as constraints results in a narrower distribution for $f$ and thus significantly reducing the radius ($\mathcal{R}_{\mathrm{HAT}}$) of the norm ball *i.e.*, $\mathcal{B}_{\mathcal{R}_{\mathrm{HAT}}} \leq \mathcal{B}_{\mathcal{R}_{\mathrm{STD}}}$. Now, we discuss the implications of the induced norm bounds on the overall regret of the algorithm.

**Corollary 1:** *Pick $\delta \in (0,1)$. Let $\beta_t(\mathcal{R}, \delta) = \mathcal{B}_\mathcal{R} + \epsilon\sqrt{2(\gamma_{t-1} + 1 + \ln\frac{1}{\delta})}$ where $\gamma_{t-1}$ is the associated information gain [Chowdhury and Gopalan, 2017] after $t-1$ rounds and $\epsilon$ is the white Gaussian noise. Let $\|f\|_k \leq \mathcal{B}_\mathcal{R}$. If $\mathcal{B}_{\mathcal{R}_{\mathrm{HAT}}} \leq \mathcal{B}_{\mathcal{R}_{\mathrm{STD}}}$, then the following holds with the probability at least $1 - \delta$,*

$$\mathcal{P}\{R_T^{HAT} \leq R_T^{STD} \; \forall T \geq 1\} \geq 1 - \delta$$

*where $R_T$ is the cumulative regret given by $R_T = \sum_{t=1}^{T} r_t$ and $r_t$ is the instantaneous regret given by $r_t = f(\mathbf{x}^*) - f(\mathbf{x}_t)$.*

**Proof:** As discussed in Lemma 1, the RKHS norm associated with a kernel function $k(\cdot, \cdot)$ is bounded within the ball of radius $\mathcal{R}$ *i.e.*, $\|f\|_k \leq \mathcal{B}_\mathcal{R}$. Further, it states that with high probability $|f(\mathbf{x}) - \mu_t(\mathbf{x})| \leq \sqrt{\beta_t}\sigma_t(\mathbf{x})$ is satisfied if $\beta_t$'s are appropriately chosen. If $|f(\mathbf{x}) - \mu_t(\mathbf{x})| \leq \sqrt{\beta_t}\sigma_t(\mathbf{x})$ holds, then the instantaneous regret is upper bounded by $2\sqrt{\beta_t}\sigma_t(\mathbf{x})$ (see Lemma 5 in Srinivas et al. [2012]). Therefore, the instantaneous regret $r_t$ is greatly influenced by the choice of $\beta_t$. By setting the value of $\beta_t$ as $\beta_t = \mathcal{B}_\mathcal{R} + \epsilon\sqrt{2(\gamma_{t-1} + 1 + \ln\frac{1}{\delta})}$ [Chowdhury and Gopalan, 2017], we can impose a tighter bounds on the cumulative regret $R_T$. The results of Theorem 1 state that, our proposed method places a tighter bound on the RKHS norm ball such that $\mathcal{B}_{\mathcal{R}_{\mathrm{HAT}}} \leq \mathcal{B}_{\mathcal{R}_{\mathrm{STD}}}$. Such restrictions forces smaller values for $\beta_t$ in our proposed method, thereby reducing the instantaneous regret $r_t$ significantly. As a consequence, the overall cumulative regret $R_T^{\mathrm{HAT}} = \sum_{t=1}^{T} r_t$ is also tightly bounded and grows at most as $\mathcal{O}\big(\mathcal{B}_{\mathcal{R}_{\mathrm{HAT}}}T\sqrt{\gamma_T} + \sqrt{T\gamma_T(\gamma_T + \ln\frac{1}{\delta})}\big)$, where $\gamma_T$ is the kernel associated maximum information gain [Srinivas et al., 2012] after $T$ iterations. Thus the result stated above follows. $\square$

## 5  Experiments

We evaluate the optimisation performance of our proposed methods on various synthetic functions and several real-world datasets. First, we validate our methods by demonstrating its sample efficiency in the global optimisation of synthetic benchmark functions. Next, we tune the hyperparameters of the Support Vector Machine (SVM) classifier operating on publicly available real-world datasets. We compare the performance against different variations of the BO algorithm. The empirical results demonstrate the superiority of our proposed framework.

## 5.1 Emulating Human Experts

The human expert is expected to know the underlying structures of the given input space. Here we assume that the expert is aware of the optimal ground truth kernel hyperparameters ($\Theta_{\mathrm{GT}}$). We are motivated from the findings in Borji and Itti [2013] that the intelligent search strategy used by humans closely resembles Bayesian optimisation. Therefore, we run a separate GP-UCB based BO algorithm with the ground truth kernel ($k_{\mathrm{GT}}$) for emulating human experts behaviour in our proposed framework. The knowledge about the (ground truth) kernel function is obtained apriori from a large number of random samples of the objective function under consideration. Further, the human experts are given an option to intervene at any stage. In this work, we have assumed that a human expert intervenes at every third iteration of the optimisation process.

We consider the following variants of standard BO and HAT based BO in our experiments.

- **HAT-RR:** Human-AI Teaming framework with recommendation correction as mentioned in Algorithm 1. The SE kernel is used as the ground truth kernel ($k_{GT}$) for this baseline.
- **HAT-RR (KFO)**: Similar to HAT-RR, except that the ground truth kernel is chosen to be the kernel tuned from a Kernel Functional Optimisation (KFO) procedure [Venkatesh et al., 2021], a more expressive form of kernel than the Squared Exponential (SE) kernel.
- **HAT-DM (KFO)**: HAT framework with the expert feedback about good and bad regions (mentioned in Algorithm 2) and the kernel being tuned with KFO procedure.
- **STD-BO**: A standard Bayesian optimisation algorithm mentioned in the supplementary material.
- **STD-BO (MOD)**: Another variant of standard Bayesian optimisation procedure but naively considers the suggestions provided by human experts in its observation model.

In the aforesaid methods, we have used SE kernel ($k_{\mathrm{SE}}$) for fitting GP surrogate models. The length-scale parameter of $k_{\mathrm{SE}}$ is tuned in the interval $[0.1, 1]$. We standardise the function output, and thus use unit signal variance for $k_{\mathrm{SE}}$. We follow the guidelines mentioned in the Kernel Functional Optimisation (KFO) framework [Venkatesh et al., 2021] for the hyperparameter selection of both the SE kernel ($\bar{k}_{\mathrm{SE}}$) and hyperkernel ($\kappa$) used in the KFO framework. The hyperparameters of $\bar{k}_{\mathrm{SE}}$ *i.e.,* the lengthscale ($\bar{l}$) and the signal variance ($\bar{\sigma}_f^2$) are tuned in the interval $[0.1, 1]$, as the bounds are always normalised. The hyperparameters $\tilde{\lambda}_h$ and $l$ of the hyperkernel $\kappa$ used in the KFO framework are tuned in the interval $(0, 1]$ and $(0, 1]$, respectively. We refer to the supplementary material for the additional details of the KFO framework used. We use $t' = d + 2$ initial observations for a $d$−dimensional problem. The threshold ($\Delta$) used in Algorithm 2 is set at $95\%$.

## 5.2 Synthetic Experiments

First, we evaluate our approach by finding the global optima of a benchmark function (Oscillator 2D, see Figure 1a). The objective function considered has varying smoothness across its input dimensions. The expert is assumed to be aware of this smoothness information (lengthscale) via a ground truth kernel *i.e.,* a kernel with the optimal hyperparameter set tuned in the light of numerous data points. The initial observations are chosen to be in the smoother regions of the objective function such that the methods have not much prior information about the input space. Comparing the kernel learning performances revealed that our approach encouraged larger lengthscale for $x_2$ and shorter lengthscale for $x_1$. The hyperparameter distribution along $x_2$ is depicted in Figure 1b.

Furthermore, we evaluate our proposed methods with the following multi-dimensional benchmark functions [Surjanovic and Bingham, 2017] with multiple local optima: **(i)** Ackley 1D, **(ii)** Gramacy & Lee 1D, **(iii)** Branin 2D, **(iv)** Oscillator 2D, **(v)** Hartmann 3D, and **(vi)** Hartmann 6D. We compare the performance of our methods against the baselines by plotting the simple regret ($\hat{r}_t^+$) given by:

$$\hat{r}_t^+ = f(\mathbf{x}^*) - \max_{\mathbf{x}_t \in \mathcal{D}_{1:t}} f(\mathbf{x}_t) \tag{6}$$

where $f(\mathbf{x}^*)$ is the true optima of the objective function. We plot the simple regret for $10 \times d + 5$ iterations, for a given $d$−dimensional problem. The convergence results obtained for the synthetic experiments after 10 random repeated runs are shown in Figure 2. It is evident from the results that our proposed method has outperformed the baselines with KFO being better than the SE kernel.

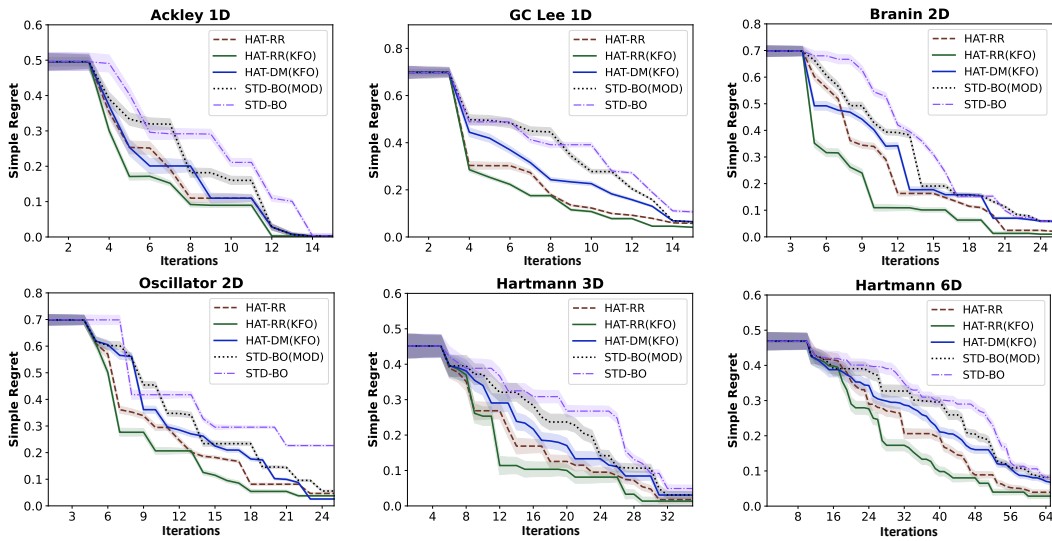

Figure 2: Simple regret vs iterations for various benchmark functions. We plot the mean regret along with its standard error obtained. The experimental results are after 10 random repeated runs.

## 5.3 Real-world Experiments

We evaluate our proposed method in tuning the hyperparameters of Support Vector Machine (SVM) classifier operating on real-world datasets. In our experiments, we consider multi-dimensional real-world datasets publicly available in the UCI repository [Dua and Graff, 2017]. The real-world datasets used are randomly split into a training set containing 80% of the total instances and a test set consisting of the remaining 20% of the total instances.

We use C-SVM with Radial Basis Function (RBF) kernel to minimise the test classification error ($E_r$). We tune the SVM hyperparameters *i.e.,* the cost parameter and the RBF kernel width in the exponent space of $[-3, 3]$ and $[-5, 1]$, respectively. The minimum classification error (in %) obtained for the test set averaged over 10 random repeated runs are shown in Table 1. We italicise the best results of our HAT-RR framework to demonstrate that our proposed method outperforms the standard Bayesian optimisation algorithm even if the ground truth kernel (squared exponential kernel) used is not very expressive of the given input space. It is even better when more expressive kernel (KFO tuned kernel) is used. Further, we believe that the superior performance of HAT-RR framework against the HAT-DM framework is because HAT-RR directly incorporates the suggestions from the human expert for function evaluation, whereas HAT-DM assesses and corrects the deviations in the current model by maximising the distance between good and bad regions.

## 5.4 Robustness to Imprecision in Human Knowledge

In addition to the experiments discussed above, we further evaluate our proposed method to show its robustness in model learning against the noisy human expert inputs. In this experiment, we emulate human errors by corrupting the human inputs by adding suitable noise to the **(i)** kernel parameters, and **(ii)** the trade-off factor $\beta_t$. In order to add noise in the kernel parameters, we use Multi Kernel Learning (MKL) [Aiolli and Donini, 2015] - a weighted combination of SE kernel ($k_{SE}$), linear kernel ($k_{LIN}$) and polynomial kernel ($k_{POL}$) *i.e.,* $k(\mathbf{x}, \mathbf{x}') = w_1\, k_{SE}(\mathbf{x}, \mathbf{x}') + w_2\, k_{LIN}(\mathbf{x}, \mathbf{x}') + w_3\, k_{POL}(\mathbf{x}, \mathbf{x}')$ for the ground truth kernel. We add a white Gaussian noise to the weights $\mathbf{w}$ *i.e.,* $w_i \sim \mathcal{N}(w_i, \hat{\sigma}^2_{GN})$ to emulate the human expert error in understanding the ground truth knowledge. To account for the human error in providing suggestions, we add noise to the trade-off factor $\beta_t$ using a Gamma distribution ($\Gamma$) *i.e.,* $\beta_t = \Gamma(\frac{\beta_t^2}{\hat{\sigma}^2_{GN}}, \frac{\hat{\sigma}^2_{GN}}{\beta_t})$. In all our experiments, we have set the noise variance $\hat{\sigma}^2_{GN} = 0.1$. We evaluate our proposed method supplemented with the imprecise human knowledge using the following synthetic functions: **(i)** Levy 2D, **(ii)** Shubert 2D, and **(iii)** Egg 2D. We compare the convergence results with the standard Bayesian optimisation algorithm. The results obtained for the aforesaid synthetic functions are shown in Figure 3. It is evident from the results

Table 1: SVM classification error rates obtained for the real-world datasets using different algorithms. Each cell value signifies the mean test classification error and its standard deviation obtained after 10 random repeated runs. Bold values indicates the best performance among all the columns. Lower the better. We italicise the best results of our HAT-RR framework to emphasise its superiority against the standard BO even when a standard unexpressive (SE kernel) is used.

| Dataset | HAT-RR | HAT-RR (KFO) | HAT-DM (KFO) | STD-BO | STD-BO (MOD) |
|---------|--------|--------------|--------------|--------|--------------|
| WDBC | $0.98\pm0.2$ | $\mathbf{0.60 \pm 0.75}$ | $0.95 \pm 0.45$ | $1.5 \pm 0.5$ | $1.24 \pm 0.2$ |
| Ionosphere | $\mathit{5.15\pm0.4}$ | $5.25 \pm 0.10$ | $6.21 \pm 0.81$ | $8.9 \pm 0.2$ | $6.02 \pm 0.6$ |
| Sonar | $6.46\pm0.3$ | $\mathbf{6.11 \pm 0.37}$ | $6.84 \pm 0.92$ | $8.21 \pm 0.9$ | $8.44 \pm 0.2$ |
| Heart | $10.2\pm0.3$ | $\mathbf{10.25 \pm 0.41}$ | $10.97 \pm 0.75$ | $11.7 \pm 0.9$ | $11.10 \pm 1.4$ |
| Seeds | $2.8\pm0.14$ | $\mathbf{2.51 \pm 0.65}$ | $2.63 \pm 0.47$ | $3.3 \pm 0.4$ | $2.58 \pm 0.8$ |
| Wine | $\mathbf{0}$ | $\mathbf{0}$ | $\mathbf{0}$ | $\mathbf{0}$ | $\mathbf{0}$ |
| Credit | $12.7\pm1.3$ | $\mathbf{12.42 \pm 0.60}$ | $12.95 \pm 0.78$ | $18.1 \pm 1.3$ | $14.52 \pm 0.6$ |
| Biodeg | $\mathit{13.4\pm0.2}$ | $13.88 \pm 0.42$ | $14.09 \pm 1.83$ | $16.8 \pm 1.9$ | $15.05 \pm 0.6$ |
| Car | $0.21\pm0.1$ | $0.35 \pm 0.17$ | $\mathbf{0.30 \pm 0.53}$ | $1.9 \pm 0.5$ | $0.39 \pm 0.7$ |
| Ecoli | $1.67\pm0.2$ | $\mathbf{1.21 \pm 0.38}$ | $1.94 \pm 0.64$ | $2.1 \pm 0.6$ | $2.01 \pm 0.3$ |

that our method performs on par with standard BO even when the expert knowledge is imprecise, as we try to incorporate expert knowledge to further improve the existing log-likelihood estimates.

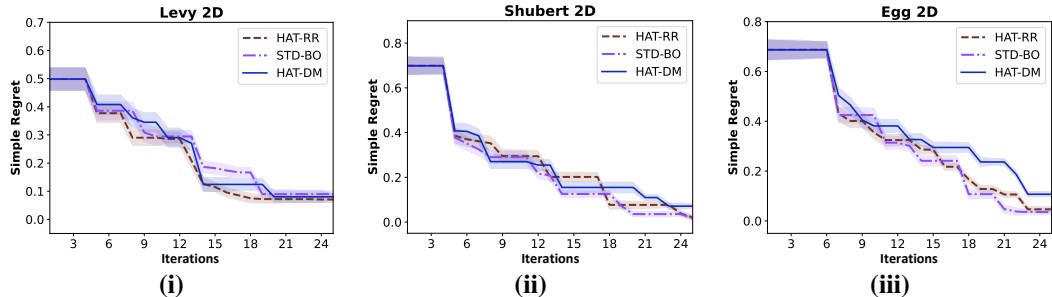

Figure 3: Simple regret plot for **(i)** Levy 2D function, **(ii)** Shubert 2D function, and **(iii)** Egg 2D function obtained using noisy HAT-RR algorithm, noisy HAT-DM algorithm, and standard BO (STD-BO). We plot the mean regret along with its standard error computed after 10 random repeated runs.

To further validate our proposed Human-AI collaborative Bayesian optimisation framework, we have conducted a study with real human experts. We have provided the additional experimental results in the supplementary material (A.3). The code base used for the experiments mentioned above is available at `https://github.com/mailtoarunkumarav/HumanAITeaming`.

## 6   Conclusion

We propose a novel Bayesian optimisation framework to accelerate the optimisation process by incorporating additional information available from human experts. We present two different expert intervention strategies. In the first strategy, we let the expert to correct the current recommendation. In the second, we seek expert's hunch on good region vs bad region for the selection of next sample based on the current observation set. We then incorporate such feedback as constraints in the model selection. We theoretically analyse our framework to show that expert's knowledge can improve sample efficiency. The experimental results show that our method outperforms other baselines.

## Acknowledgements

This research was partially funded by Australian Government through Australian Research Council (ARC). Prof. Venkatesh is the recipient of an ARC Australian Laureate Fellowship (FL170100006).

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
