## A.1 Bayesian Optimisation

Bayes theorem states that, given the data $\mathcal{D}$ and the model $\mathcal{M}$, the posterior probability of the model given data $\mathcal{P}(\mathcal{M}|\mathcal{D})$ is directly proportional to the likelihood of the data given model *i.e.,* $\mathcal{P}(\mathcal{D}|\mathcal{M})$, multiplied by the prior probability of model $\mathcal{P}(\mathcal{M})$,

$$\mathcal{P}(\mathcal{M}|\mathcal{D}) \propto \mathcal{P}(\mathcal{D}|\mathcal{M})\,\mathcal{P}(\mathcal{M}) \tag{7}$$

If $y_t = f(\mathbf{x}_t) + \epsilon_t$ denotes a noisy observation of the unknown objective function $f$ for the $t^{\text{th}}$ sample $\mathbf{x}_t$ corrupted with white Gaussian noise ($\epsilon_t$), then the observation model $\mathcal{D}$ is accumulated as $\mathcal{D}_{1:t} = \{\mathbf{x}_{1:t}, \mathbf{y}_{1:t}\}$. Bayesian optimisation computes the posterior distribution $\mathcal{P}(f\,|\mathcal{D}_{1:t})$ by combining the prior $\mathcal{P}(f)$ with the likelihood $\mathcal{P}(\mathcal{D}_{1:t}|\,f)$ as,

$$\mathcal{P}(f\,|\mathcal{D}_{1:t}) \propto \mathcal{P}(\mathcal{D}_{1:t}|\,f)\,\mathcal{P}(f) \tag{8}$$

The obtained posterior distribution represents our updated belief about the unknown objective function being modelled. There are two main aspects that must be taken into account for the Bayesian optimisation. First, the selection of priors to express our prior belief about the function being modelled. Gaussian processes are used in defining prior distributions for the unknown objective function. Second, the acquisition function to determine the next promising point for the function evaluation. Therefore, Bayesian optimisation consists of two main components: **(i)** a Gaussian process model [Williams and Rasmussen, 2006] of $f$, and **(ii)** acquisition functions [Kushner, 1964] to guide the search for optima.

### A.1.1 Acquisition Functions

The predictive distribution from GP is used to select the next query point with the high promise of finding the optima. The selection of the next best query point is characterised by an acquisition function $u(\mathbf{x})$. The acquisition function guides the search for the optimum by balancing exploration of high variance regions versus exploitation of high mean regions in the input space. At all levels of our experiments, we have used Upper Confidence Bound (UCB) selection criterion for the acquisition function. Gaussian Process-Upper Confidence Bound (GP-UCB) acquisition function [Srinivas et al., 2012] using the UCB selection criterion is given as follows.

$$u_{\text{GP-UCB}}(\mathbf{x}) = \mu(\mathbf{x}) + \sqrt{\beta_t}\,\sigma(\mathbf{x})$$

where $\beta_t$ is a trade-off parameter that balances the exploration-exploitation. In this research work, we follow Chowdhury and Gopalan [2017] to appropriately set the values for $\beta_t$. Chowdhury and Gopalan [2017] have discussed in detail the convergence analysis of a variant of GP-UCB acquisition function to demonstrate its superiority when compared to other acquisition functions. Kushner [1964] proposed Expected Improvement (EI) acquisition function to guide the search for optima by taking into account the expected improvement over the current known optima. The next best query point is obtained by maximising the EI acquisition function $u_{\text{EI}}(\mathbf{x})$, given by,

$$u_{\text{EI}}(\mathbf{x}) = \begin{cases} (\mu(\mathbf{x}) - f(\mathbf{x}^+))\,\Phi(Z) + \sigma(\mathbf{x})\,\phi(Z) & \text{if } \sigma(\mathbf{x}) > 0 \\ 0 & \text{if } \sigma(\mathbf{x}) = 0 \end{cases} \tag{9}$$

$$Z = \frac{\mu(\mathbf{x}) - f(\mathbf{x}^+)}{\sigma(\mathbf{x})}$$

where $\Phi(Z)$ and $\phi(Z)$ represents the CDF and PDF of the standard normal distribution, respectively and $f(\mathbf{x}^+)$ is the best value observed so far.

The standard Bayesian optimisation procedure is as mentioned in Algorithm 3.

---

**Algorithm 3** Standard Bayesian Optimisation

---

**Input**: Set of observations $\mathcal{D}_{1:t'} = \{\mathbf{x}_{1:t'}, \mathbf{y}_{1:t'}\}$, Sampling budget $T$

    1. **for** $t = t', \cdots, T$ iterations **do**

    2.        Optimise hyperparameters: $\Theta^* = \underset{\Theta}{\operatorname{argmax}} \log \mathcal{L}$

    3.        Update GP model with $\Theta^*$

    4.        Find the next query point $\mathbf{y}_{t+1} = \underset{\mathbf{x} \in \mathbb{X}}{\operatorname{argmax}} u(\mathbf{x})$

    5.        Query the objective function $f(\mathbf{x})$ at $\mathbf{x}_{t+1}$ as $y_{t+1} = f(\mathbf{x}_{t+1}) + \epsilon_{t+1}$

    6.        Augment the data as $\mathcal{D}_{1:t+1} = \mathcal{D}_{1:t} \cup (\mathbf{x}_{t+1}, y_{t+1})$

    7.        Update GP model with $\mathcal{D}$

    8. **end for**

---

## A.2 Theoretical Proofs

The formal definition of Sobolev spaces is as mentioned below.

**Definition 1** ([Tartar, 2007]): *For a non-negative integer* $\mathfrak{m}$*, for* $1 \leq P \leq \infty$ *and for a compact set* $\omega \subset \mathbb{R}^d$*, the Sobolev space* $W^{\mathfrak{m},P}(\omega)$ *is the space of functions* $\psi \in L^P(\omega)$ *such that the derivatives* $D^\tau \psi \in L^P(\omega)$ *for all derivatives of length* $|\tau| \leq P$*. It is a normed space equipped with the norm given as* $\|\psi\| = \sum_{|\tau| \leq \mathfrak{m}} \|D^\tau \psi\|_P$*, or the equivalent norm given as:*

$$\|\psi\|_{\mathfrak{m},P} = \left( \int_\omega \Big( \sum_{|\tau| \leq \mathfrak{m}} |D^\tau \psi|^P \Big) dx \right)^{\frac{1}{P}} \qquad if \ 1 \leq P \leq \infty$$

$$\|\psi\|_{\mathfrak{m},\infty} = \underset{|\tau| \leq \mathfrak{m}}{\operatorname{argmax}} \|D^\tau \psi\|_\infty \qquad if \ P = \infty$$

A special case of Sobolev space with $P = 2$ *i.e.,* $W^{\mathfrak{m},2}$ is a Sobolev Hilbert space.

**Proof of Theorem 1:** We prove Theorem 1 using the notion of Sobolev Hilbert spaces [Tartar, 2007]. With kernel function $k(\cdot, \cdot)$ on $\mathbb{R}^d$ and $\mathbb{X} \subset \mathbb{R}^d$, we represent the RKHS of $k$ on Lipschitz continuous domain $\mathbb{X}$ as $\mathcal{H}(\mathbb{X})$. Now, we follow Bull [2011] to provide a characterisation of $\mathcal{H}_{k_{\Theta_{\text{HAT}}}}(\mathbb{X})$ by defining $\mathcal{H}_{k_{\Theta_{\text{HAT}}}}(\mathbb{R}^d)$. Let $\hat{f}$ be the Fourier transforms for the functions $f$ in $L^2$ space and $\hat{k}$ be the Fourier transforms for the continuous and integrable kernel functions $k$. We use the following lemma from Bull [2011] to generally define any RKHS $\mathcal{H}(\mathbb{X})$ in terms of $\mathcal{H}(\mathbb{R}^d)$.

**Lemma 2**: *Let* $\mathcal{H}(\mathbb{R}^d)$ *be the space of real continuous functions* $f_k \in L^2(\mathbb{R}^d)$ *whose norm*

$$\|f_k\|^2_{\mathcal{H}(\mathbb{R}^d)} := \int \frac{|\hat{f}(\xi)|}{\hat{k}(\xi)} \, d\xi$$

*is finite.* $\mathcal{H}(\mathbb{X})$ *is the space of functions* $f_k = g|_{\mathbb{X}}$ *for some* $g \in \mathcal{H}(\mathbb{R}^d)$ *with norm* $\|f_k\|_{\mathcal{H}(\mathbb{X})} := \underset{g|_{\mathbb{X}} = f_k}{\inf} \|g\|_{\mathcal{H}(\mathbb{R}^d)}$.

As discussed in Tartar [2007] the above mentioned spaces correspond to the Sobolev Hilbert spaces if we assume a Lipschitz continuous domain $\mathbb{X}$. Formally, the Sobolev Hilbert space $\mathcal{H}^s(\mathbb{X})$ is the space of functions $f : \mathbb{X} \to \mathbb{R}$, with some restriction on $g : \mathbb{R}^d \to \mathbb{R}$ and the finite norm (as per Lemma 2) represented as:

$$\|f\|^2_{\mathcal{H}^s(\mathbb{X})} := \underset{g|_{\mathbb{X}} = f}{\inf} \int \frac{|\hat{g}(\xi)|}{\hat{k}(\xi)} \, d\xi \tag{10}$$

With suitable assignments for the Fourier transforms [Bull, 2011] given in Eq. (10), we get the restricted RKHS $\mathcal{H}(\mathbb{X})$. Now we see such restricted function space mapped in $g$ as the Hilbert space containing the narrower distribution of $f$ i.e., $\mathcal{H}^s_{k_{\Theta_{HAT}}}$ enforced by the constraints used in the HAT framework. As our proposed HAT framework deals with a restricted RKHS similar to that of the Sobolev Hilbert spaces, we use the RKHS norm formulations defined in the Sobolev Hilbert spaces to define upper bounds on the induced RKHS norm of the HAT framework. The induced RKHS norm in the Sobolev Hilbert space satisfies the inequality $\|f\|^2_{\mathcal{H}^s(\mathbb{X})} \leq \|g\|^2_{\mathcal{H}(\mathbb{R}^d)}$. In standard BO, the complete distribution of the objective function is used in the modelling process, whereas the HAT framework restricts some parts of the distribution of $f$ on the verge of satisfying the additional constraints added in the modelling process. With the formulations and inequalities mentioned above, we say that the RKHS norm ($\|f\|^2_{\mathcal{H}^s(\mathbb{X})}$) for HAT framework is not greater than that of the RKHS norm of the standard BO i.e., $\|f\|^2_{\mathcal{H}^s_{k_{HAT}}} \leq \|f\|^2_{\mathcal{H}_{k_{STD}}}$. Furthermore, with the relationship established on the induced RKHS norm, we say that the radius ($\mathcal{R}_{HAT}$) of the norm ball obtained in our proposed framework is significantly smaller than the norm ball radius ($\mathcal{R}_{STD}$) of the standard approach without any expert involvement i.e., $\mathcal{B}_{\mathcal{R}_{HAT}} \leq \mathcal{B}_{\mathcal{R}_{STD}}$, concluding the proof of Theorem 1. $\qquad\square$

## A.3 Experimental Details and Additional Results

### A.3.1 Parameter Selection in Kernel Functional Optimisation

We follow the guidelines mentioned in [Venkatesh et al., 2021] to set the algorithmic parameters of KFO framework. We use Matérn Harmonic Hyperkernel to define the kernel functional space. To construct kernels as kernel functionals in Hyper-RKHS, we consider an evenly spaced grid of size $N_g \gtrsim 10n$ for a given $n$ dimensional problem. The outer-loop ($S$) of KFO representing the number of low-dimensional subspace searches to find the best kernel functional is restricted to $S = 5$ and the inner-loop i.e., the number of iterations ($T$) in each of the subspace $s \in S$ is restricted to $T = 10$.

### A.3.2 Analysis of Synthetic Experimental Results

It is important to note that the algorithms do not converge do the same regret value, it is either because of the scaling factor in plotting or due to the termination before converging (exhausted evaluation budget). Further, we would like to recall here that BO is guaranteed to reach zero regret only asymptotically [Srinivas et al., 2012]. Hence, with a finite number of iterations, we are never guaranteed to reach zero regret. Therefore, generally, we look at the optimiser's ability to converge as close to the global optima as feasible given a limited evaluation budget. Most of the state-of-the-art optimisers will converge to the global optima if a sufficient evaluation budget is allocated. However, ensuring a larger evaluation budget is not feasible, especially in the optimisation of expensive black-box objective functions. Therefore, rather than noting the final optimal regret attained, we evaluate the performance of the optimiser by looking at its sample-efficiency i.e., how good are the solutions obtained at each step/iteration.

Based on the empirical results mentioned in Figure 2 (mentioned in the main paper), we summarise the ability of our framework to converge to better solutions ($\lambda\%$ close to the global optima) in Table 2. We have reported the average evaluation budget (along with its standard error) consumed to reach $\lambda\%$ close to the global optima for the synthetic functions using our Human-AI Teaming (HAT) framework and the standard BO (STD-BO) procedure. As observed from Table 2, for Ackley function, our method takes approximately 5 iterations to reach $40\%$ close to global optima, whereas the standard BO procedure consumed 11 iterations to provide similar solutions, thus making our proposed HAT framework much more sample-efficient. Furthermore, using the descriptive statistics from Table 2, we computed $99.7\%$ confidence intervals ($[\mu + 3\sigma, \mu - 3\sigma]$) for both HAT-RR-KFO and STD-BO to verify the statistical difference in their distributions. We note that the distributions do not match and the differences in simple regret reduction are statistically significant.

Table 2: Average number of iterations (along with its standard error) required for the HAT framework and STD-BO to arrive at $\lambda\%$ close to the global optima.

| $\lambda\%$ | Method | Ack-1D | Brn-2D | Osc-2D | Har-3D | Har-6D |
|---|---|---|---|---|---|---|
| 40% | HAT-RR-KFO | $4.7 \pm 0.03$ | $7.4 \pm 0.07$ | $6.9 \pm 0.07$ | $11.4 \pm 0.1$ | $26.9 \pm 0.1$ |
| | STD-BO | $11.1 \pm 0.1$ | $15.3 \pm 0.1$ | $20.2 \pm 0.0$ | $26.7 \pm 0.7$ | $53.3 \pm 0.1$ |
| 20% | HAT-RR-KFO | $7.8 \pm 0.02$ | $9.7 \pm 0.06$ | $13.8 \pm 0.04$ | $20.5 \pm 0.10$ | $41.6 \pm 0.1$ |
| | STD-BO | $13.0 \pm 0.2$ | $20.2 \pm 0.0$ | $33.1 \pm 0.0$ | $30.9 \pm 0.0$ | $62.6 \pm 0.1$ |
| 10% | HAT-RR-KFO | $11.4 \pm 0.02$ | $16.8 \pm 0.02$ | $24.1 \pm 0.01$ | $26.9 \pm 0.07$ | $51.6 \pm 0.1$ |
| | STD-BO | $13.5 \pm 0.02$ | $23.2 \pm 0.02$ | $46.2 \pm 0.02$ | $42.7 \pm 0.04$ | $76.8 \pm 0.1$ |
| 1% | HAT-RR-KFO | $11.9 \pm 0.02$ | $31.8 \pm 0.01$ | $38.7 \pm 0.01$ | $36.8 \pm 0.02$ | $74.4 \pm 0.05$ |
| | STD-BO | $13.9 \pm 0.02$ | $46.1 \pm 0.03$ | $71.8 \pm 0.01$ | $58.8 \pm 0.03$ | $91.6 \pm 0.5$ |

### A.3.3 Descriptive Statistics of Real-world Datasets

The descriptive statistics of the real-world datasets used from the UCI repository are mentioned in Table 3. We refer to the UCI website for the detailed explanation of the attributes and its characteristics. The datasets used are randomly split into a training set containing $80\%$ of the total instances and a test set consisting of the remaining $20\%$ of the total instances.

Table 3: Descriptive statistics of the publicly available real-world datasets [Dua and Graff, 2017] used in the SVM classification experiments.

| Dataset | Features | Samples | Classes | Missing Values | Attribute Characteristics |
|---|---|---|---|---|---|
| WDBC | 30 | 569 | 3 | None | Real |
| Ionosphere | 34 | 351 | 2 | None | Integer, Real |
| Sonar | 60 | 208 | 2 | None | Real |
| Heart | 13 | 303 | 2 | Yes | Categorical, Integer, Real |
| Seeds | 7 | 210 | 3 | None | Real |
| Wine | 13 | 178 | 3 | None | Real |
| Credit | 24 | 1000 | 2 | Yes | Categorical, Integer |
| Biodeg | 41 | 1056 | 3 | None | Integer, Real |
| Car | 6 | 1728 | 4 | None | Categorical |
| Ecoli | 8 | 314 | 8 | None | Real |

### A.3.4 Experiments with Real Human Expert

To further validate our framework, we have conducted a small study with real human experts. We have considered the function optimisation of the same Oscillator 2D function (mentioned in Figure 1a of the main paper) for the experiments with a real human expert. The contour map of the objective function considered is as shown in Figure 4a. As seen in the contour map, the objective function is rough along $x_1$ and smooth along $x_2$. To record the suggestions of a human expert, we have designed a 2-dimensional user interface (see Figure 4b) that shows the current set of observations and also the AI model suggestion in the current iteration. Each point indicated by a red dot corresponds to an observation ($\mathbf{x}$) in $\mathcal{D}_{1:t}$ and the associated annotations corresponds to the noisy function value $y = f(\mathbf{x}) + \epsilon$, where $\epsilon$ is the white Gaussian noise. The recommendation given by AI system in the current iteration is denoted by blue triangle in Figure 4b.

We have conducted two experiments with two different human experts (Expert1 and Expert2) having varying degrees of exploration-exploitation strategy. In the first experiment, the expert (Expert1) considered was more of exploiting nature, whereas in the second experiment the expert (Expert2) was keen on exploring the given input space, rather than exploiting the known best solution. In both the experiments, the expert is given the information about the smoothness of the objective function *i.e.,* the objective function has sharp variations along $x_1$ and is smooth along $x_2$. Using such information about the smoothness of the objective function under consideration, we believe that a human expert can come up with meaningful suggestions.

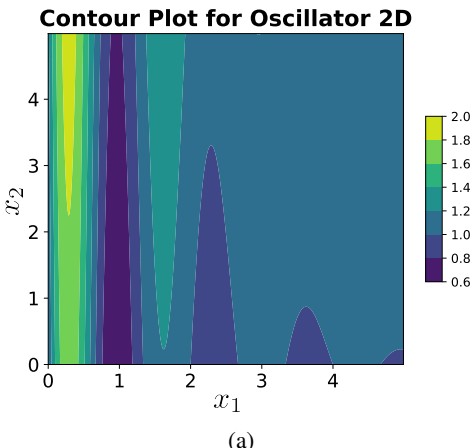
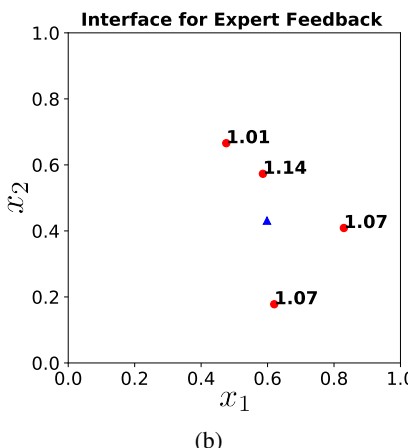

|   |   |
|---|---|
| (a) | (b) |

Figure 4: **(a)** Contour map of Oscillator $2-$dimensional benchmark function **(b)** A $2-$dimensional user interface to record human experimenter's feedback, where each red dot correponds to a observation ($\mathbf{x}$) in $\mathcal{D}$ and the annotations correspond to the function value ($y$) observed for each $\mathbf{x}$. Blue triangle corresponds to the AI suggestion in the current iteration.

In this experiment, we have considered 4 initial observations and run the optimisation procedure for 20 iterations. We have considered the recommendation rectification strategy (HAT-RR) mentioned in Algorithm 1 (mentioned in the main paper) for this study. We have compared our method involving a human expert (HAT-RR-Expert) against the standard Bayesian optimisation (STD-BO) algorithm mentioned in Algorithm 3. We plot simple regret ($\hat{r}_t^+$) mentioned in Eq. (6) of the main paper to measure the optimisation performance of the methods considered. The results obtained for the experiments involving 2 different human experts (Expert1 and Expert2) are shown in Figure 5a and 5b respectively. It is observed from the results of the first experiment (see Figure 5a) that the expert tried to exploit the known best solution and seldom explores the input space. In contrast, the results from the second experiment revealed that the expert was able to slightly explore around the input space and thus, relatively slower convergence. In both the experiments, the baseline (STD-BO) struggled to converge and kept exploring the input space due to the wider distribution of the function space resulting in sub-optimal performance. Our small-scale real-world study demonstrated that the human expert feedback can be proven useful in boosting the optimisation performance significantly.

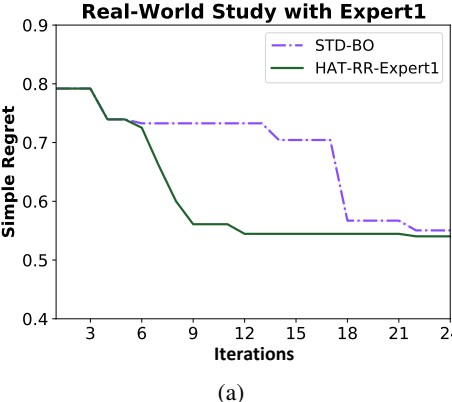
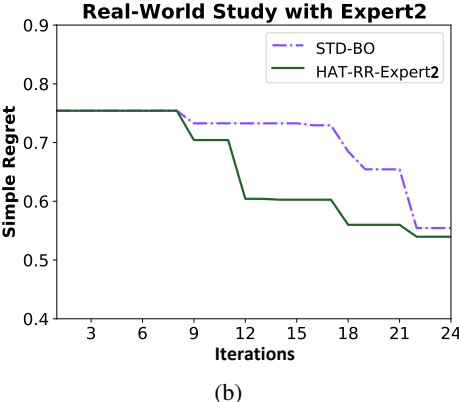

|   |   |
|---|---|
| (a) | (b) |

Figure 5: Simple regret plotted for **(a)** the first experiment with a real human expert (Expert1) following more of exploitation strategy, and **(b)** the second experiment with a real human expert (Expert2) following more of exploration strategy.