# OpenReview forum: "Human-AI Collaborative Bayesian Optimisation"
_NeurIPS.cc/2022/Conference — NeurIPS 2022 Accept_

### Official Review · Reviewer_R7wD · 2022-07-10

**Rating:** 6
**Confidence:** 4
**Soundness:** 2 fair
**Presentation:** 2 fair
**Contribution:** 3 good

**Summary:**

This paper proposes a framework that makes use of expert's knowledge in improving the effectiveness of Bayesian optimization. The Human-AI Teaming (HAT) based Bayesian optimization framework takes expert inputs as constraints via the following two ways:
- Rectifying recommendations by making use of the correlation structure of the experimental system by updating the hyperparameters with the constraints that the acquisition function of the human model is better than the one of the AI model.
- Specifying good and bad regions based on the current observation set by updating the hyperparameters that optimize the difference between the acquisition function of the good regions and the bad regions and achieve at least \Delta of the original likelihood.
Furthermore, the paper also provides theoretical analysis that shows additional information from human experts enforced as constraints result in a narrower distribution, thus improving sample efficiency.

Finally, the effectiveness of the proposed framework is tested using both synthetic data and real-world experiment of tuning the hyperparameters of SVM.

**Questions:**

In the synthetic experiment section, Figure 2 shows the simple regret for 10*d+5 iterations for the different models. Why choose "10" here?  It seems that after certain iterations, although the human AI teamed method converge faster (i.e., big drops in simple regret in early iterations than standard BO procedure), they reach the optimal simple regret almost at the same iteration as the standard BO procedure. For example, Ackley 1D, human AI teamed model reached 0.0 regret around iteration 13 and standard BO around iteration 14. Since the iteration is capped at "10*d+5", it's not clear if human-AI will get to the global optimal faster than the standard BO procedure. The experiments does show that when the number of iterations (e.g. computation budget) is limited, using a human-AI teamed BO method will give better results.

Typos:
- Line 278: "Algorithm 2" should be "Algorithm 1"

**Limitations:**

The proposed method will improve the performance of Bayesian optimization method when the computation budget is limited; however, it does not seem to find the optimal solution in significant fewer iterations. One improvement to add in the experiment section is to provide a figure of the number of iterations of the optimal regret (after the optimizer is converged) for all the comparing models.

**Strengths And Weaknesses:**

Strength:
- This paper is well written and flows well.
- This paper provides theoretical analysis of the proposed method.
- This paper provides an original idea of incorporating human experts knowledge into the Bayesian optimization procedure through constraitns
- The effectiveness of the proposed method is demonstrated in both synthetic and real-world experiments

Weakness:
- It's not clear whether the proposed method will achieve the optimal solution faster than the standard Bayesian optimization based on the presented results.

---

> ### Author Response · Authors · 2022-08-02
> **Response to reviewer R7wD (Part 1 of 2)**
>
>
> We thank our reviewer's effort to raise important points and provide valuable suggestions.
>
> - **Comment:** _“Simple regret for $10d+5$ iterations for the different models. Why choose "$10$" here?”_
> **Reply:**
>
>
>     - We have drawn inspiration from the other literature [1,2,3,4] in the BO research community to use this rule-of-thumb in setting the evaluation budget. We have validated the allocated budget with our experimental results. In most of the synthetic experiments, our method has reached optimal (or at least near-optimal) solutions after $10\times d+5$ iterations for a $d$ dimensional problem.
>
> ___
>
> - **Comment:** _“Whether the proposed method will achieve the optimal solution faster than the standard Bayesian optimization?” it does not seem to find the optimal solution in significant fewer iterations.?_
> **Reply:**
>
>     - We would like to recall here that BO is guaranteed to reach zero regret only asymptotically [5]. Hence, with a finite number of iterations we are never guaranteed to reach zero regret. Therefore, generally we look at the optimiser's ability to converge as close to the global optima as feasible given a limited evaluation budget.
>
>     - Most of the state-of-the-art optimisers will converge to the global optima if a sufficient evaluation budget is allocated. However, ensuring a larger evaluation budget is not feasible, especially in the optimisation of expensive black-box objective functions. Therefore, rather than looking at the optimal regret attained, we evaluate the performance of the optimiser by looking at its sample efficiency _i.e.,_ how good are the solutions at each iteration.
>
>     - Table 1 (see below) signifies the ability of our framework to converge to better solutions ($x$% close to the global optima). We have reported the average evaluation budget (along with its standard error) consumed to reach $x$% close to the global optima for the synthetic functions using our Human-AI Teaming (HAT) framework and the standard BO (STD-BO) procedure.
>
>     - As observed from Table 1, for Ackley function, our method takes approximately $5$ iterations to reach $40$% close to global optima, whereas the standard BO procedure consumed $11$ iterations to provide similar solutions, thus making our proposed BO framework much more sample-efficient.
>
>     - Furthermore, using the descriptive statistics from Table 1, we have computed $99.7$% confidence intervals $([\mu+3\sigma, \mu-3\sigma])$ for both HAT-RR-KFO and STD-BO to verify the statistical difference in their distributions. We observed that the distributions do not match and the differences in simple regret reduction are statistically significant.
>
> ___
> - **Comment:** _“Experiments: Figure of the number of iterations of the optimal regret (after the optimizer is converged) for all the comparing models.”_
> **Reply:**
>
>     - We agree with our reviewer's comment to add a metric that signifies the optimiser's ability to reach optimal regret in the Experiments section.
>
>     - As mentioned earlier, BO is assumed to reach optimal zero regret asymptotically. In addition to the metric mentioned in Table 1, we also provide the average number of iterations (along with its standard error) to reach $1$% close to the global optima. It is evident from the table that our framework outperforms the standard BO procedure in converging as close to the global optima as feasible.
>
> **Table 1**: Number of iterations required for the HAT framework and std-BO to arrive at $x$% close to the global optima.
>
>       |        |       40% close       |         20% close     |       10% close       |
>       ----------------------------------------------------------------------------------
>       |        | HAT-RR-KFO|   STD-BO  | HAT-RR-KFO|   STD-BO  | HAT-RR-KFO|  STD-BO   |
>       |---------------------------------------------------------------------------------
>       | Ack-1D |  4.7±0.03 | 11.1±0.04 |  7.8±0.02 | 13.0±0.02 | 11.4±0.02 | 13.5±0.02 |
>       | Brn-2D |  7.4±0.07 | 15.3±0.04 |  9.7±0.06 | 20.2±0.02 | 16.8±0.03 | 23.2±0.02 |
>       | Osc-2D |  6.9±0.07 | 20.2±0.03 | 13.8±0.04 | 33.1±0.02 | 24.1±0.01 | 46.2±0.02 |
>       | Har-3D | 11.4±0.15 | 26.7±0.07 | 20.5±0.10 | 30.9±0.06 | 26.9±0.07 | 42.7±0.04 |
>       | Har-6D | 26.9±0.13 | 53.3±0.17 | 41.6±0.09 | 62.6±0.10 | 51.6±0.07 | 76.8±0.10 |
>
>
> **Table 2**: Number of iterations required for the HAT framework and std-BO to arrive at $1$% close to the global optima.
>
>       |            |   Ackley  |  Branin   |   Osc-2D  |  Hart-3D  |  Hart-6D  |
>       |-------------------------------------------------------------------------
>       | HAT-RR-KFO | 11.9±0.02 | 31.8±0.01 | 38.7±0.01 | 36.8±0.02 | 74.4±0.05 |
>       | STD-BO     | 13.9±0.02 | 46.1±0.03 | 71.8±0.01 | 58.8±0.03 | 91.6±0.05 |
> ___
>
> _Due to the space constraints, we have provided references in the following comment: **Response to reviewer R7wD (Part 2 of 2)**_

---

> > ### Comment · Reviewer_R7wD · 2022-08-09
> > **Response**
> >
> > I thank the reviewer for their thorough response. My concern has been adequately addressed. It'll make the empirical evidence much stronger if Table 1 and Table 2 can be added to the main body of the paper.

---

> > > ### Author Response · Authors · 2022-08-09
> > > **Thanks for the acknowledgment**
> > >
> > > We thank the reviewer for reading and acknowledging our rebuttal. We will incorporate the reviewer suggestions on additional tabular results in the main body of the paper.

---

> ### Author Response · Authors · 2022-08-02
> **Response to reviewer R7wD (Part 2 of 2)**
>
> _Please refer to Part1 of our responses in the comment **Response to Reviewer R7wD (Part 1 of 2)**_
>
> **References**
>
> 1. Ha, Huong, Santu Rana, Sunil Gupta, Thanh Nguyen, and Svetha Venkatesh. "Bayesian optimisation with unknown search space." Advances in Neural Information Processing Systems 32 (2019).
>
> 2. Hutter, Frank, Holger Hoos, and Kevin Leyton-Brown. "An evaluation of sequential model-based optimisation for expensive blackbox functions." In Proceedings of the 15th annual conference companion on Genetic and evolutionary computation, pp. 1209-1216. 2013.
>
> 3. Liu, Jingfei, Chao Jiang, and Jing Zheng. "Batch Bayesian optimisation via adaptive local search." Applied Intelligence 51, no. 3 (2021): 1280-1295. Harvard.
>
> 4. Souza, Artur, Luigi Nardi, Leonardo B. Oliveira, Kunle Olukotun, Marius Lindauer, and Frank Hutter. "Bayesian Optimisation with a Prior for the Optimum." In Joint European Conference on Machine Learning and Knowledge Discovery in Databases, pp. 265-296. Springer, Cham, 2021.
>
> 5. N. Srinivas, A. Krause, S. M. Kakade, and M. W. Seeger. Information-theoretic regret bounds for Gaussian process optimisation in the bandit setting. IEEE Transactions on Information Theory, 58(5):3250–3265, 2012.

---

### Official Review · Reviewer_dMaE · 2022-07-11

**Rating:** 6
**Confidence:** 2
**Soundness:** 2 fair
**Presentation:** 3 good
**Contribution:** 2 fair

**Summary:**

The paper proposes a new method for human-AI collaboration in the context of Bayesian optimization, where the optimization algorithm can get occasionally help from a human expert with deeper knowledge about the underlying physical phenomenon, either in the form of a correction of the current recommendation within the search space or specifying good and bad regions within the search space. Authors provide a theoretical framework justifying their approach and an empirical validation of the method against a baseline method without expert feedback.


**Questions:**

- Could you better explain the statement at line 29, expanding more on why the nature of application ensure that there would be an expert running the optimization process?
- Are there other works in the literature introducing human feedback into Bayesian optimization or optimization processes? I would expand the background section by mentioning the state of the art of works including the human in the loop of optimization processes.
- How would authors explain the better performance of HAT with recommendation correction than with bad/good regions feedback? (see Fig.2)
- In real-world experiments: which was the classification task? And, how was the feedback from humans provided in this case?

**Limitations:**

Authors don't address limitations of their work.
The main limitation of including a feedback expert in the optimization process is that the performance relies heavily on the human true level of "experience" about the process under analysis, and besides such feedback goodness varies from expert and expert. I suggest authors to mention their view about this and other limitations as well as improvement perspectives, in the Conclusion section.

**Strengths And Weaknesses:**

The paper contribution is clearly presented and its effectiveness in accelerating the optimization process is supported theoretically and validated both using synthetic and real world data. The  paper is articulated in a clear way and the overall quality is fair.
The background section and results discussion do not report other works about collaborative human-AI in optimization processes, therefore an assessment of the paper's contribution originality and significance in the context of related work is hard.

---

> ### Author Response · Authors · 2022-08-02
> **Response to Reviewer dMaE (Part 1 of 2)**
>
> We appreciate our reviewer's time and effort to provide a thoughtful review.
>
> - **Comment:** _“why the nature of application ensure that there would be an expert running the optimization process?”_
> **Reply:**
>
>     - In the experimental design optimisation process, the main goal is to iteratively design, perform experiments and eventually converge to the optimal design using a minimal number of trials. For e.g., in material designing, if the objective is to cast a particular alloy with high tensile strength, then a material scientist has to iteratively cast a series of materials with different compositions to eventually reach a composition with target strength.
>
>     - Bayesian optimisation (BO) provides a principled data-driven framework for the experimental optimisation of black-box and expensive design processes. BO has been widely applied in design optimisation applications [1,2] pertaining to material manufacturing, aerospace engineering and robotics.
>     - In each iteration of the design-experiment loop, BO suggests a design that is highly likely to be the optimum design. Then the design suggested by BO is evaluated by an expert to verify its optimality. Next, the experimental results are fed back into BO to improve the inherent machine learning models, such that it recommends better designs in the upcoming iterations in the light of data.
> ___
> - **Comment:** _“Are there other works in the literature introducing human feedback into Bayesian optimization?”_
> **Reply:**
>
>     - We consider the reviewer's suggestion in expanding the background section to include the details of the prior art on incorporating human feedback into BO.
>
>     - In most of the prior art, the expert knowledge being incorporated is static and usually specified at the beginning of the optimisation process. There have been some early attempts to incorporate such static human feedback into the BO process.
>
>         - [3] proposed to incorporate the static monotonicity information directly into the Gaussian Process (GP) via virtual observations.
>         - [4] proposed to incorporate shape constraints in BO by restricting partial derivative processes.
>         - [5] proposed to incorporate the unimodality assumptions by adding a set of virtual derivative observations in GP.
>         - [6] proposed to optimise functions that are not directly accessible for evaluation by incorporating the pair-wise preferences from human experts.
>         - [7] proposed to incorporate the shape of functions in the GP modelling via spatially varying kernels.
>     - Methods that incorporate static properties are problem-specific. However, here we assume fluid form of knowledge where expert improves her knowledge on the shape of the functions as the experiments progress.
> ___
> - **Comment:** _“Better performance of HAT with recommendation correction than with bad/good regions feedback?”_
> **Reply:**
>
>     - We believe that the better performance of HAT-RR (recommendation rectification) vs. HAT-DM (good/bad regions) is because HAT-RR directly incorporates the suggestions from the human expert for function evaluation, whereas HAT-DM assesses and corrects the deviations in the current model by maximising the distance between good and bad regions.
> ___
> - **Comment:** _“Which was the classification task? and how was the feedback from humans provided in this case?”_
> **Reply:**
>     - In real-world experiments, we use the proposed collaborative BO to tune the hyperparameters of C-SVM performing binary classification tasks on real-world datasets from the UCI data repository [8]. For example, in real-world experiments with _WDBC_ dataset, we tune the hyperparameters (i.e., cost $C$ and kernel width $\gamma$) of C-SVM using our framework and classify the given test instances into _“Benign or Malignant”_. Similarly, with _Biodeg_ dataset, we improve the the C-SVM accuracy in classifying biodegradable materials by tuning its hyperparameters.
>
>     - The prior art [9] concludes that the strategy used by humans for active search closely resembles BO with some exploration-exploitation trade-off. Inspired by such findings, we emulate human experts by running a separate BO procedure with its inherent GP models constructed using an accurate ground truth kernel signifying the expert's (full or partial) knowledge about the correlation structures. The predictive GP distribution is then used to find the next best candidate for the function evaluation by maximising the acquisition function. This candidate point suggested is used as human feedback in our algorithm.
>
>     - In our experiments with C-SVM, we emulate a human expert by running a separate BO optimising the cost and kernel width parameter of C-SVM. The candidate SVM hyperparameters suggested by the emulated expert is incorporated as human feedback in our framework.
> ___
>
> _Due to the space constraints, we have provided our remaining responses in the following comment: **Response to Reviewer dMaE (Part 2 of 2)**_.

---

> ### Author Response · Authors · 2022-08-02
> **Response to Reviewer dMaE (Part 2 of 2)**
>
> _Please refer to Part1 of our responses in the comment **Response to Reviewer dMaE (Part 1 of 2)**_
>
> - **Comment:** _“Limitations of the proposed framework”_
> **Reply:**
>
>     - We agree with the reviewer's comment on rewriting the conclusion section to provide the limitations of our work and discuss the future line of research.
>
>     - Motivated from the prior art [9] we assumed that a human expert uses a form of BO, though this may be generalised to an extent but may not be acceptable in all the scenarios. Although our experimental results prove the robustness of our proposed approach against possible noise in the strategy, we did not consider if human experts could use a non-BO strategy e.g., either pure exploration or pure exploitation strategy.
>
>     - Our framework does not capture the confidence levels of the human expert in suggesting the candidate points, that may possibly contain useful information. Currently, we assume that the experts intervene when they are confident in suggesting candidate points. However, experts can still reveal some useful information even when they are not very confident.
>
>     - Current human expert based study is limited in number of participants and thus can only provide indicative results.
>
> ___
>
> **References**
>
> 1. Rubín de Celis Leal, David, Dang Nguyen, Pratibha Vellanki, Cheng Li, Santu Rana, Nathan Thompson, Sunil Gupta et al. "Efficient Bayesian Function Optimisation of Evolving Material Manufacturing Processes." ACS omega 4, no. 24 (2019): 20571-20578.
>
> 2. Oliveira, Rafael, Lionel Ott, and Fabio Ramos. "Bayesian optimisation under uncertain inputs." In The 22nd international conference on artificial intelligence and statistics, pp. 1177-1184. PMLR, 2019. Harvard
>
> 3. Li, Cheng, Rana Santu, Sunil Gupta, Vu Nguyen, Svetha Venkatesh, Alessandra Sutti, David Rubin et al. "Accelerating Experimental Design by Incorporating Experimenter Hunches." In 2018 IEEE International Conference on Data Mining (ICDM), pp. 257-266. IEEE Computer Society, 2018.
>
> 4. Michael Jauch and Victor Peña. Bayesian optimisation with shape constraints. arXiv preprint arXiv:1612.08915, 2016
>
> 5. Michael Riis Andersen, Eero Siivola, and Aki Vehtari. Bayesian optimisation of unimodal functions. In Neural Information Process Systems (NeurIPS) workshop on Bayesian optimisation, 2017.
>
> 6. Javier González, Zhenwen Dai, Andreas Damianou, and Neil D Lawrence. Preferential Bayesian optimisation. In International Conference on Machine Learning, pages 1282–1291. PMLR, 2017.
>
> 7. Arun Kumar, A. V., Santu Rana, Cheng Li, Sunil Gupta, Alistair Shilton, and Svetha Venkatesh. "Bayesian Optimisation for Objective Functions with Varying Smoothness." In Australasian Joint Conference on Artificial Intelligence, pp. 460-472. Springer, Cham, 2019.
>
> 8. D. Dua and C. Graff. UCI machine learning repository, 2017. URL http://archive.ics. uci.edu/ml.
>
> 9. A. Borji and L. Itti. Bayesian optimisation explains human active search. Advances in neural information processing systems, 26:55–63, 2013.

---

> ### Comment · Reviewer_dMaE · 2022-08-09
> **Response**
>
> I thank the authors for their comments. They clarified the value of their work contribution compared to the state of the art, and they provided supplementary results supporting the usefulness of their approach, showing that their method is faster than the standard BO procedure in converging to the global optima. Authors also described how to consider the effect of a human expert wrong suggestion on the algorithm’s performance. I agree with reviewer 7Niz about the fact that a proper human study would improve the paper’s quality. However, I think changes introduced by the authors made the contribution clearer to me.

---

> > ### Author Response · Authors · 2022-08-10
> > **Thanks for the acknowledgement**
> >
> > We thank our reviewer for acknowledging and carefully reading through our rebuttal to understand our contributions.

---

### Official Review · Reviewer_7Niz · 2022-07-18

**Rating:** 6
**Confidence:** 3
**Soundness:** 2 fair
**Presentation:** 3 good
**Contribution:** 2 fair

**Summary:**

This paper proposes to augment Bayesian optimization (Gaussian processes based) formulation with human experts intervention on suggesting sample points (different from the Bayesian optimization suggestion) and human's good and bad sample regions (to be exact they are sample points instead of regions as it is formulated in the paper now).  The introduction of human intervention into BO is an interesting and potentially useful contribution. However, the nature of human intervention is not rigorous enough missing  formal  characterization of the exact requirement on human experts and practical verification of such human properties in human study. For example, we might need verifiable formal notion of what if human expert is wrong, what kind of wrongness, and how wrong the system can endure a human expert but still guarantee a bound of the optimality as well as the sampling process efficiency.


**Questions:**

- what if human expert is wrong?
- what kind of wrongness, and how wrong the system can endure a human expert but still guarantee a bound of the optimality as well as the sampling process efficiency?

**Limitations:**

Please improve the study with (1) a formal  characterization of the exact requirement on human experts and practical verification of such human properties in human study; (2) non-Gaussian processes BO formulation such as: Garnelo, Marta, et al. "Neural processes." arXiv preprint arXiv:1807.01622 (2018), and its follow-up work.


**Strengths And Weaknesses:**

Strength
	- The introduction of human expert recommended samples as well as good and bad region suggestions into the BO formulation

Weakness
	- Experiments lack real human experts human study
	- No formal treatment of human experts properties and how these properties will be factored into the proposed augmented BO
	- The BO formalization will be better off if it can also handle non-Gaussian processes BO formulation such as: Garnelo, Marta, et al. "Neural processes." arXiv preprint arXiv:1807.01622 (2018), and its follow-up work.

---

> ### Author Response · Authors · 2022-08-02
> **Response to reviewer 7Niz**
>
>
> We appreciate the reviewer's valuable comments and overall feedback.
>
> - **Comment**: _“Formal characterization of the exact requirement on human experts and practical verification of such human properties in human study”_
> **Reply:**
>     - In this work, we expect the human expert to be aware of the underlying correlation structures of the objective function, but not the optima.
>
>     - For example, an expert may already know that the objective function is smoother over dimension $X_{1}$ than $X_{2}$. Now, when the expert sees Bayesian Optimisation (BO) recommendation, she may notice that BO has not figured out the above-mentioned information, and thus may be recommending samples that are equally close to existing samples. She then may correct the sample by pushing the sample way further out on $X_{1}$ than $X_{2}$. Such corrections significantly improve the convergence rates.
>
>     - In practice [1], a human expert is assumed to use a form of BO strategy. The experts are expected to be aware of the underlying correlation structures via ground truth kernel ($k_{\text{GT}}$) having ground truth hyperparameters ($\theta_{\text{GT}}$). Therefore, we run a separate BO algorithm with $k_{\text{GT}}$ and $\theta_{\text{GT}}$ to emulate human experts.
>
>     - We refer to the supplementary material (Section **A.3.1**) for our experiments with two human experts that practically verified the knowledge requirements of human experts. In this real-world user study, we considered two categories of experts:
>         - Exploring expert: an expert who tries to explore the given input space and sample around unexplored regions.
>         - Exploiting expert: an expert who tries to exploit the known best solutions and play safely in consuming evaluation budget.
> ___
>
> - **Comment**: _“What if human expert is wrong? what kind of wrongness, and how wrong the system?”_.
> **Reply:**
>
>     - The level of consistency and accuracy in the human expert knowledge significantly varies from expert to expert. We discuss the following two cases describing the wrongness of the expert input.
>
>     - **Case 1**: The expert is slightly wrong - We have performed experiments (Section **A.3.2** in the supplementary material) where we have slightly perturbed the ground truth kernel ($k_{\text{GT}}$) and the GP-UCB trade-off parameter  $(\beta_{t})$ to simulate randomness associated with human decisions. Our experimental results demonstrated that even with imprecise human knowledge, HAT-BO can still provide competitive results.
>
>     - **Case 2**: The expert is totally wrong e.g. kernel ($k_{\text{GT}}$) is wrong or trade-off parameters are off - then the human corrected samples will be less useful. But any sample would add information to the GP and thus BO should be able to exploit that and keep optimising sub-linearly in the iterations where expert does not intervene. In that extreme case where expert intervenes all the time, we may not achieve convergence. So it is important that expert intervention is done mostly during the beginning.
> ___
>
> - **Comment**:_“BO formalization will be better off if it can also handle non-Gaussian processes such as Neural Processes”_
> **Reply:**
>     - We thank our reviewer for the suggestion. Neural Processes [2] can be used for building surrogate models in our proposed framework provided we uncover ways in which human knowledge can be absorbed in such non-kernel-based surrogates.
>
>
>
> **References**
>
> 1. A. Borji and L. Itti. Bayesian optimization explains human active search. Advances in neural information processing systems, 26:55–63, 2013.
>
> 2. Garnelo, Marta, Jonathan Schwarz, Dan Rosenbaum, Fabio Viola, Danilo J. Rezende, S. M. Eslami, and Yee Whye Teh. "Neural processes." arXiv preprint arXiv:1807.01622 (2018).

---

> > ### Comment · Reviewer_7Niz · 2022-08-08
> > **response**
> >
> > It will be better if there is a human study to confirm that human experts are really a form BO algorithms with some better intuitiion of a form of ground truths as characterized by the emulated human experts in the paper; also human experts can improve the BO in real sense. Otherwise, it would be more like a boosting schema in ML to create stronger classifiers/regressors out of week classifiers/regressors --- creating stronger BOs out of  multiple weeker BOs (weaker or of different properties). As a reslt, this would be less of human-AI collaboration but more about another BO algorithmic schemes.

---

> > > ### Author Response · Authors · 2022-08-09
> > > **Response to Reviewer 7Niz on the emulated experts**
> > >
> > > We thank our reviewer's time and effort to read through our rebuttal and provide valuable feedback.
> > >
> > > As per [1] we know that active human search mimics BO. We also know that BO involves two dominant aspects:
> > > 1. Estimating the GP model _i.e.,_ the kernel function.
> > > 2. Building the acquisition function.
> > >
> > > Here, Step 2 is a more intrinsic step on how humans search, whereas Step 1 will dictate how rapidly the search converges. Whilst a non-expert needs to do both the steps, an expert (our model of expert) needs to just perform Step 2. In short, we believe that knowing only the kernel function won't be enough to alter the nature of the search. It might just accelerate the convergence by making an expert behave as if it is few steps ahead of a non-expert. This is our reasoning on emulating an expert using BO with GP using the ground-truth kernel function.
> > >
> > > **References**
> > >
> > > 1. A. Borji and L. Itti. Bayesian optimisation explains human active search. Advances in neural information processing systems, 26:55–63, 2013.

---

### Meta-Review · Area_Chair_c1kS · 2022-08-26

**Recommendation:** Accept
**Confidence:** Less certain

**Metareview:**

Reviewers found the contribution of introducing human feedback into Bayesian optimization novel, interesting, and sound. (It is worth noting that the author rebuttal was crucial in assuaging several reviewer concerns and confusions.) A more extensive human study would significantly strengthen the work, but that said, the paper should be an interesting contribution to NeurIPS as is.

**Award:**

No

---

### Decision · Program_Chairs · 2022-09-14

Accept